# Concordance between Pressure Platform and Pedigraph

**DOI:** 10.3390/diagnostics11122322

**Published:** 2021-12-09

**Authors:** Cristina Gonzalez-Martin, Uxia Fernandez-Lopez, Abian Mosquera-Fernandez, Vanesa Balboa-Barreiro, Maria Teresa Garcia-Rodriguez, Rocio Seijo-Bestilleiro, Raquel Veiga-Seijo

**Affiliations:** 1Research Group in Nursing and Health Care, Instituto de Investigación Biomédica de A Coruña (INIBIC), Complexo Hospitalario Universitario de A Coruña (CHUAC), Sergas, Universidade da Coruña (UDC), 15071 A Coruña, Spain; vanesa.balboa.barreiro@sergas.es (V.B.-B.); maria.teresa.garcia.rodriguez@sergas.es (M.T.G.-R.); rocio.seijo.bestilleiro@sergas.es (R.S.-B.); raquel.veiga.seijo@udc.es (R.V.-S.); 2Departament of Health Sciences, Faculty of Nursing and Podiatry, Campus of Esteiro, University of A Coruña, 15471 Ferrol, Spain; m.uxia.fernandez.lopez@udc.es (U.F.-L.); abian@udc.es (A.M.-F.)

**Keywords:** pedigraph, pressure platform, foot

## Abstract

Objectives: Determine the concordance between two methods of obtaining the plantar footprint (pedigraph and pressure platform). Methods: A descriptive, cross-sectional, observational study of prevalence was carried out in the social center of Cariño (Coruña), Spain (*n =* 65 participants). Older people without amputations or the presence of dysmetria were included. The variables studied were: sociodemographic (age, sex), anthropometric (body mass index) and footprint measurement variables. These measurements were made by obtaining the plantar footprint using two methods: pedigraph and pressure platform. Results: The mean age of the sample was 37.42 ± 15.05 years, with a predominance of the female gender (61.54%). Positive linear correlation between pedigraph and platform was observed in both feet in the Chippaux and Staheli indices (correlation coefficient > 0.3, *p* < 0.001 in each comparison). The reliability was good or moderate in relation with the Chippaux and Staheli index. Slightly lower coefficients were observed in the dimensions of the foot. Conclusions: A positive linear correlation between pedigraph and platform was observed in both feet in the Chippaux and Staheli indices. Significant differences were observed between pedigraph and platform in relation to the width and length of the foot. It is probably due to the fact that the pressure platform provides more exhaustive, detailed and accurate information of the foot.

## 1. Introduction

The function of the human foot is influenced by its anatomical structure. The shape of the plantar arch and its main support points in the heel and metatarsal area allow the weight of the body to be supported without the foot sinking [1,2]. The functional and structural characteristics of the foot vary with many factors, such as age, sex, weight, the presence of systemic diseases (such as diabetes or other comorbidities) [3], the fact of practicing a sports technique [4,5] and genetic disorders, such as Down’s syndrome in which muscle, ligament problems and gait disturbances appear [6]. These circumstances promote the need to assess the lower limb in a systematic, individualized and detailed way, including different techniques that allow studying the foot with rigor and quality.

Among these techniques, it is worth highlighting the analysis of the plantar footprint as a widely studied method, defined as simple and valid for studying the foot [7]. Its analysis will provide information to classify the foot according to its characteristics, and also to decide on possible treatments, in line with the study of other parameters [8]. It should be noted that the plantar print is the image of the surface of the foot that contacts the ground, and that the different morphologies of the internal longitudinal arch can be transferred to the plantar print with different characteristics. For this reason, its analysis can constitute a complementary method in the exploration and diagnosis of its morphology, in addition to the fact that it is known that the height of the arch modifies the pressure distribution pattern on the plantar surface [9]. For this reason, these methods are used in addition to other screening and assessment.

Among the plantar footprint capture methods, the manuals, such as the pedigraph or the pressure platform, stand out. However, although there are studies that have analyzed the concordance of the plantar footprint assessment indices [10,11], no studies have been found that tried to see the concordance of both methods of obtaining the plantar footprint by studying the different indices and angles described in the literature, such as the Staheli index, the Chippaux–Smirak index, or the Clarke Angle [12].

Therefore, the objective of this research work is to determine the concordance between two methods of obtaining the plantar footprint (pedigraph and pressure platform).

## 2. Materials and Methods

### 2.1. Participants

A total of 65 subjects were analyzed, with a mean age of 37.42 ± 15.05 years, with a predominance of the female gender (61.54%). In total, 49.23% of the studied sample was overweight or obese.

### 2.2. Design and Scope of the Study

A descriptive, cross-sectional, observational study of prevalence was carried out in the social center of Cariño (Coruña), Spain. During the research period, *n* = 65 (α = 0.05; β = 0.20) participants were studied following a consecutive sampling method. A sample size of 18 participants allowed us to estimate a Pearson correlation coefficient r = 0.62 with a 95% certainty and a statistical power of 80%, in a two-sided contrast. Data collection was carried out in May and June 2021.

### 2.3. Inclusion and Exclusion Criteria of the Studied Sample

People of legal age (>18 years) who wanted to participate in this study were included in the study. Those people who presented with dysmetria and amputations of the lower limb were excluded and those who did not sign the informed consent.

### 2.4. Variables Studied and Procedure

The variables included in this study were: sociodemographic (age, sex), anthropometric (body mass index), footprint measurement variables and foot width and length (cm). These measurements were made by obtaining the plantar footprint using two methods: pedigraph and pressure platform. In this way, two plantar prints of each foot per subject were obtained through the mentioned methods. Different measurements were made in each footprint, from different angles and indices described in the literature: Chippaux–Smirak index and Staheli index. Each of these parameters is used to categorize the footprint as cavus, normal or flat.

Through the pedigraph (Laboratorios Herbitas S.L., Valencia, Spain) (Figure 1), the footprint was obtained by stepping on a rubber device impregnated in ink, under which there is a blank DIN A4 paper on which the person’s footprint is transferred. To obtain an accurate footprint and in conditions similar to the other method of obtaining the footprint, the subject was initially seated and the pedigraph was placed under the foot and then the person stood, allowing the weight of the body to fall.

The second method of obtaining the plantar footprint was through a pressure platform (Sensor Medica SRL, Rome, Italy; controller S/N: 130-220000132). The person was placed on the pressure platform, and after assessing the appropriate position, the plantar footprint was recorded in static. An example of this can be seen in Figure 2.

The study of the plantar footprints was carried out through three measurements (Figure 3) [11], which were the following:Chippaus–Smirak index. It was evaluated by dividing a line that joins the narrowest area of the isthmus and a parallel line in the widest area of the forefoot. This result is multiplied by 100, since this index is expressed as %. The normal range is 35 ± 10%, values greater than 45% will be cavus feet and less than 25% will be flat feet.Staheli index. It was obtained by dividing the narrowest part of the isthmus by the value of one parallel at the widest part of the heel. The values described to assess it are given because they are the normal range of 0.6 to 0.69, values greater than 0.69 will be pes cavus and less than 0.6 will be flat feet.

The performance of the footprints through the different methods and the corresponding measurements were carried out by two podologists previously trained for this purpose.

### 2.5. Ethical-Legal Aspects

This study was approved by the Galicia Research Ethics Committee (CEIG 2019/079). In addition, the ethical and legal aspects were fulfilled and followed in this investigation, respecting the confidentiality of the data. All subjects were invited to participate in this research on a voluntary basis, being informed of the objective of the study and what their participation consisted of. All the subjects signed the informed consent to participate in this research.

### 2.6. Statistic Analysis

A descriptive analysis of the variables under study was performed; the continuous variables are described as mean ± SD, median, range and interquartile range. The qualitative variables were expressed as absolute and relative frequency.

Pearson’s correlation coefficients and their *p*-values were provided for the correlation analysis between the measurements of both instruments used (pedigraph and platform).

The estimates of the inter-class correlation coefficients (ICC) and their 95% confidence intervals were calculated, as described by Shrout and Fleiss [13]. All calculations were performed using the absolute agreement and single rater method, with two-way random effects ICC (2.1). The reliability values of the ICC were classified as poor (<0.50), moderate (0.50–0.75), good (0.75–0.90) and excellent (>0.90) [14].

Bland–Altman plots were constructed to determine whether there were fixed biases or substantial outliers between the two measurement techniques, with the difference of the two measurements for each sample on the vertical axis and the average of the two measurements on the horizontal axis. Statistical analysis was performed using the SPSS v 24.0 statistical package.

## 3. Results

A total of 65 subjects were analyzed, with a mean age of 37.42 ± 15.05 years, with a predominance of the female gender (61.54%). In total, 49.23% of the studied sample was overweight or obese (Table 1).

Four footprint measurements were analyzed by pedigraph and platform in both lower limbs (LF and RF), in addition to the forefoot and hindfoot load through the platform (Table 1). A clear predominance of normal footprint was observed according to the Chippaux index measured by the pedigraph. Measurements of this index through the platform continued to provide a higher percentage of normal footprint, with a more discreet prevalence. Regarding the Staheli index, the pediatrician classified around 50% of the feet as cava footprint, while this percentage increased when making measurements with the platform.

A positive linear correlation was observed between the measurements obtained by pedigraph and platform in the Chippaux and Staheli indices (correlation coefficient > 0.3, *p* < 0.001 in both indices and according to foot) (Table 2). There was no correlation between both instruments in terms of foot width or foot length.

The analysis of variance determined, in most cases, the absence of bias between the measures provided by the two instruments, observing non-significant differences between the two devices. Significant differences were only observed between pedigraph and platform in the Staheli index of the left lower limb and foot length (LF and RF). The intraclass correlation coefficients according to a random effects model of the variance of two factors (Table 3) indicate that the reproducibility (reliability) of the average of the measures obtained with both instruments is good or moderate (ICC = (0.5–0.9)) in the case of the Chippaux and Staheli index, with coefficients ranging between 0.694 and 0.825. Slightly lower coefficients were observed in the dimensions of the foot. The lowest reproducibility was obtained in the width of the left foot, classified as poor (ICC = 0.292, *p* = 0.233). The magnitude of the differences in the measurements of both instruments can be seen in Figure 1. In this figure, we find that very few measurements have agreed (difference equal to zero). In the Staheli index and the length of the foot (Figure 4C,D,G,H), a clear predominance of positive differences is appreciated, indicating that the pedigraph provides higher values than the platform. On the other hand, it is the pedigraph that provides the highest values, in terms of the Chippaux index (Figure 4A,B) and the width of the foot (Figure 4E,F).

The concordance between the pedigraph and platform methods for the diagnosis of flat, cavus or normal foot in the total sample and according to foot, is shown in Table 4. A greater similar concordance was observed in both indices and in the width and length of the foot.

## 4. Discussion

The main objective of this study was to show the concordance between two methods of footprint assessment, such as pedigraphy and pressure platform. Even though both assessment methods are widely known and used, to the best of our knowledge, few projects have been carried out to study the concordance between them. The parameters that are subject to study in this research have not been included in the same way by other researchers, as will be noted in this section.

First of all, it is important to highlight that the use of foot measurements to classify the morphology of the foot remains one of the focal concepts of lower-extremity biomechanics [15]. Research that has looked at different methods of studying the height of the internal longitudinal arch shows mixed results concerning the best method for determining this parameter [16]. In addition, although some studies compare different parameters in pedigraphy and the pressure platform, the state of the literature is much more extensive in terms of research that has compared these measurements in the same method. For this reason, a brief discussion of these studies has also been added to this section.

In this way, some articles have been found that study concordance in the classification of the morphology of the footprint, and the results obtained are varied [11,17]. Thus, the study carried out by Diéguez Varela [18], in addition to determining the validity of the Staheli index and the Chippaux–Smirak index taking the Arch index as a reference, also established the intra-observer agreement of these indices, reaching the conclusion that they are reliable indices to be used in the determination of the plantar footprint and observing that the Staheli index has greater agreement than the Chippaux–Smirak index. On the other hand, González-Martin et al. [11] studied the concordance between the Clark angle and the Chippaux–Smirak and Staheli indices in renal transplant patients, with the Chippaux–Smirak index obtaining the highest level of concordance. The results can be justified by the difference in measurement of these indices, as the Chippaux–Smirak index measures the midfoot while the Staheli index measures the rearfoot.

Other authors studied other footprint parameters to evaluate the correlation among them [19]. For example, Zuil-Escobar et al. [19] found out that the correlation between the navicular drop test and the footprint parameters evaluated were good. However, the authors pointed out that the navicular drop test had fewer disadvantages than using footprint parameters. Instead, González-Martín et al. [20] pointed out that Clarke’s angle has limited sensitivity in diagnosing flat feet, using the Chippaux–Smirak index as a reference. Furthermore, it is necessary to consider that other parameters, such as BMI values, also affect all of the assessment.

### 4.1. Reliability and Accuracy

Regarding the predominant footprint type in our research, both the pedigraph and the pressure platform show a predominance of the normal foot according to the Chippaux–Smirak index, while the Staheli index shows a predominance of the pes cavus. Other researchers who studied these parameters and others found out excellent intrarater reliability [21].

Concerning the accuracy of pedigraph and pressure platform and the footprint parameters evaluated, the statistical analysis reported no statistical differences in the parameters studied. We found differences in the width of the foot. Despite this, other authors [21] have not reported differences in the parameters studied, although they did not study the same parameters as this research (for example, they did not include foot width or forefoot and rearfoot loading). Other authors studied other parameters, such as the longbow angle, for which no significant differences were found either [22].

On the other hand, it is also necessary to show the research of Fascione, Crews and Wrobel [23]. They evaluated the differences between different parameters, including the Staheli index and the Chippaux–Smirak index, using different methods, among which were pedigraphy and the pressure platform, finding significant differences, which differs from our results and those of Zuil-Escobar et al. [21].

For this reason, this research was the first to assess the accuracy of the different parameters by pedigraph and platform pressure in static. Thus, although the differences between the two methods of obtaining the footprint are not significant for the two indices studied, it is necessary to add that it is now known that the pressure platform provides more exhaustive, detailed and accurate information on foot loading than the pedigraph. The presence of sensors along the platform gives more accurate data compared to the traditional method [24]. This reason supports the poor agreement and reliability, and is due to the sensors along the platform pressure system and the improvement of technologies in this type of system. An example of this according to the results of this work is related to the measurement of the forefoot.

Furthermore, the additional measurements provided by the pressure platform in conjunction with the more accurate sensing of the pressure exerted by the foot under load means that we currently favor this method. This is because the combination of position and pressure data gives a more accurate footprint.

The practical and clinical applicability of the results of this work is that it is necessary to perform a complete and comprehensive examination of all persons on an individual basis. The use of the different measurements studied, such as the Chippaux–Smirak index and the Staheli index, are complementary to the assessment to be carried out. The pressure platform presents a technological system that provides more information on the pressure system of the foot on the ground.

This work invites further research on clinical assessment methods in routine practice, to continue adding parameters for biomechanical assessment based on scientific evidence.

### 4.2. Limitations

Among the limitations of our study, it is necessary to point out that not all parameters of the plantar footprint have been considered. The most widely used in recent years have been studied. In addition, the sample studied presents different types of feet, including normal, flat and cavus. It may be necessary to compare these parameters on a homogeneous foot type. Finally, this work has only been carried out in the adult population aged between 18 and 65 years, not including children or older people, so we can only generalize the results to this age group.

## 5. Conclusions

The normal footprint was the most prevalent footprint according to the Chippaux index, while the dug footprint was the most prevalent according to the Staheli index.The frequency of the normal footprint according to the Chippaux index classification was higher in the measurements provided by the pedigrapher compared to those provided by the platform.The Staheli index detected a higher percentage of cavus footprint through the platform.A positive linear correlation was observed between pedigrapher and platform in relation to the measurements taken to classify the foot according to the type of footprint.A poor agreement was observed between the two measuring instruments, especially in relation to the assessment of the width and length of the foot.The poor agreement is due to the fact that the pressure platform provides more exhaustive, detailed and accurate information on foot loading than the pedigraph.

Poor concordance was observed between both measurement instruments.

## Figures and Tables

**Figure 1 diagnostics-11-02322-f001:**
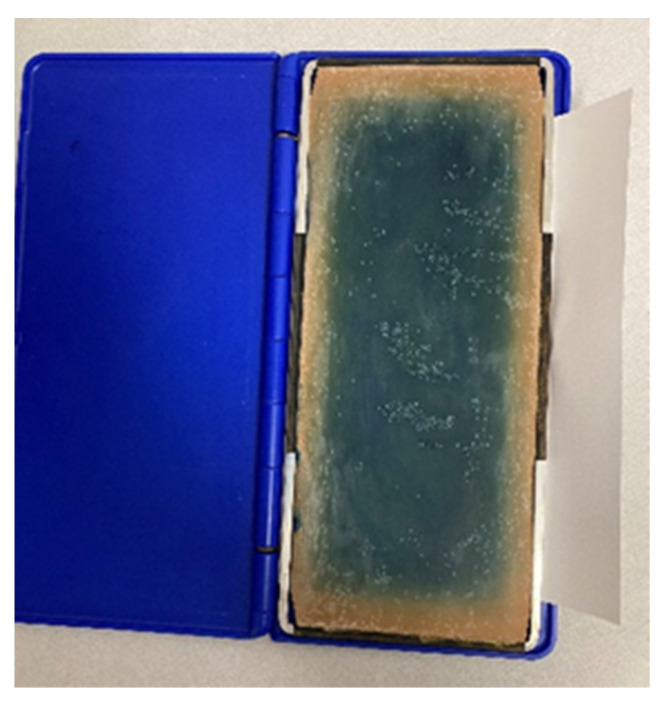
Pedigraph.

**Figure 2 diagnostics-11-02322-f002:**
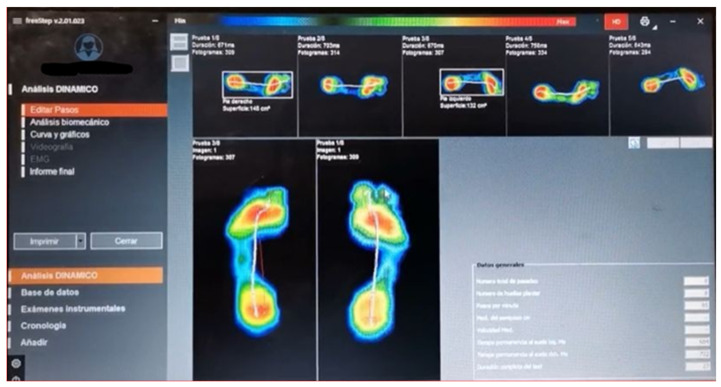
Study of the plantar footprint using the pressure platform.

**Figure 3 diagnostics-11-02322-f003:**
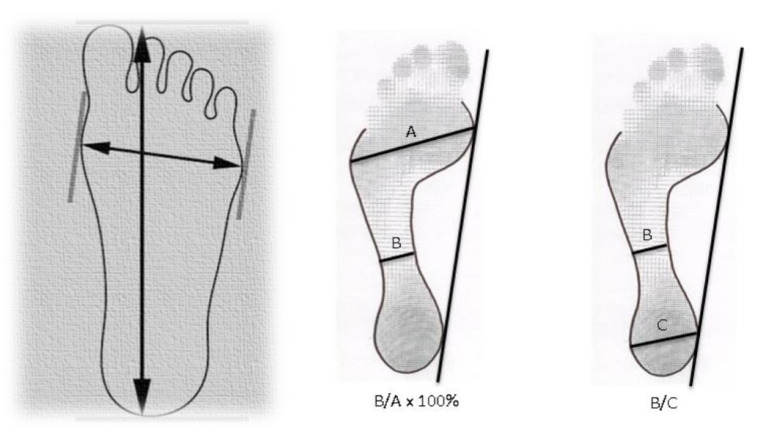
Measurements made on the footprints. (**A**) Length and width of the foot; (**B**) Chippaux–Smirak index; (**C**) Staheli index.

**Figure 4 diagnostics-11-02322-f004:**
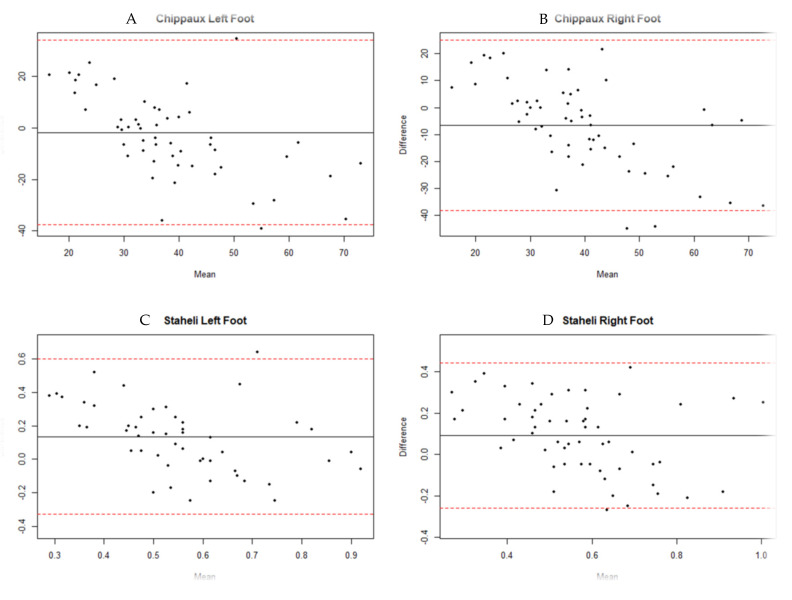
Bland–Altman analysis plots. The central solid line represents the mean difference. The upper and lower solid lines represent the 95% confidence interval for the limits of agreement (mean of difference ± 1.96 standard deviation of difference). (**A**): Chippaux Index Left Foot, (**B**): Chippaux Index Right Foot, (**C**): Staheli Index Left Foot, (**D**): Staheli Index Right Foot, (**E**): Left foot width, (**F**): Right foot width, (**G**): Left Foot length, (**H**): Right Foot length.

**Table 1 diagnostics-11-02322-t001:** Characteristics of participants (*n* = 18).

	*n*	%	Mean	Median	SD	Min	Max	Q1	Q3
**Gender**									
Female	40	61.54							
Male	25	38.46							
**Age (years)**	65		37.42	31	15.05	14	76	26	49
**Height (cm)**	65		164.68	165	11.28	118	193	160	172
**Weight (kg)**	65		72.06	69	18.56	41	130	55	85
**BMI (kg/m^2^)**									
**Normal (BMI < 25)**	33	50.77							
**Overweight/Obese (BMI ≥ 25)**	32	49.23							
**Podiatric footprints measurements**									
**Pedigraph**									
Chippaux index (LF)	64		36.88	34.35	11.51	18.4	83.7	29.85	39.85
Chippaux index (RF)	65		36.74	35	12.36	19.5	83.3	28.7	39.8
Staheli index (LF)	64		0.61	0.59	0.16	0.37	1.34	0.51	0.66
Staheli index (RF)	65		0.62	0.57	0.17	0.36	1.13	0.51	0.67
Foot width (LF) (cm)	65		8.59	8.5	0.77	6.9	10.4	8	9
Foot width (RF) (cm)	65		8.43	8.4	1.32	0.2	10.5	8	9.2
Foot length (LF) (cm)	65		22.65	23	2.08	18.5	28	21.5	24
Foot length (RF) (cm)	65		22.65	23	2	18.5	27.5	21.2	23.5
**Platform pressure**									
Chippaux index (LF)	54		39.85	36.95	18.82	6.25	88	30	50
Chippaux index (RF)	56		42.88	40.1	18.88	10.9	90.9	30.3	53.35
Staheli index (LF)	54		0.49	0.46	0.21	0.1	0.95	0.36	0.62
Staheli index (RF)	56		0.53	0.52	0.21	0.12	1	0.38	0.69
Foot width (LF) (cm)	65		6	5	2.23	2.7	11.4	4.5	8.1
Foot width (RF) (cm)	65		6.03	5	2.25	2.7	12	4.5	8
Foot length (LF) (cm)	65		14.66	12	5.27	6.3	24.3	11.1	20.2
Foot length (RF) (cm)	65		14.61	12	5.24	6.5	24	11.2	21
Forefoot load (LF)	65		25.2	25	3.34	18	33	23	27
Forefoot load (RF)	65		26.85	27	4.45	16	37	24	30
Retropie load (LF)	65		24.58	24	4	18	34	22	27
Retropie load (RF)	65		23.82	24	3.86	14	34	21	26

BMI: body mass index. LF: left foot. RF: right foot. SD: standard deviation. Min: minimum. Max: maximum. Q1: the first quartile. Q3: the third quartile.

**Table 2 diagnostics-11-02322-t002:** Correlation between podiatric measurements for both raters of pedigraph and platform pressure.

	Pearson’s Correlation Coefficient	*p*	*n*
**Chippaux Index**			
Total (both feet)	0.339	<0.001	109
LF	0.395	0.003	54
RF	0.445	0.001	56
**Staheli Index**			
Total (both feet)	0.374	<0.001	110
LF	0.28	0.04	54
RF	0.511	<0.001	56
**Foot Width (cm)**			
Total (both feet)	0.13	0.139	130
LF	0.162	0.197	65
RF	0.182	0.147	65
**Foot Length (cm)**			
Total (both feet)	0.011	0.903	130
LF	0.044	0.73	65
RF	0.039	0.756	65

LF: left foot. RF: right foot. *p*: *p*-value. *n* = number of subjects.

**Table 3 diagnostics-11-02322-t003:** Interclass correlations coefficients (ICC) for two raters (pedigraph and platform pressure).

	ICC (2.1)	CI 95%		*p*	Classification
**Chippaux Index**					
Total (both feet)	0.423	0.167	0.602	0.001	poor
LF	0.496	0.13	0.708	0.007	poor
RF	0.566	0.26	0.746	<0.001	moderate
**Staheli Index**					
Total (both feet)	0.469	0.176	0.652	<0.001	poor
LF	0.306	−0.107	0.577	0.05	poor
RF	0.631	0.325	0.793	<0.001	moderate
**Foot Width (cm)**					
Total (both feet)	0.097	−0.113	0.288	0.122	poor
LF	0.081	−0.156	0.312	0.231	poor
RF	0.115	−0.173	0.369	0.186	poor
**Foot length (cm)**					
Total (both feet)	0.005	−0.102	0.125	0.467	poor
LF	0.008	−0.142	0.184	0.46	poor
RF	0.001	−0.146	0.175	0.494	poor

LF: left foot. RF: right foot. ICC: interclass correlation coefficient. CI: confidence interval. *p*: *p*-value.

**Table 4 diagnostics-11-02322-t004:** Concordance between the Chippaux and Staheli index according to foot.

		Footprint by Platform			
	**Footprint by pedigraph**				
**Chippaux Index**					
		**Cavus**	**Normal**	**Flat**	
**Total (both feet)**	**Cavus**	2 (50.0)	1 (25.0)	1 (25.0)	
	**Normal**	12 (13.3)	49 (54.4)	29 (32.2)	
	**Flat**	3 (20.0)	3 (20.0)	9 (60.0)	
		**Kappa index**	**CI 95%**		
	**Concordance**	0.173	0.001	0.345	poor
	**Observed agreement**	55%			
**Left foot**		**Cavus**	**Normal**	**Flat**	
	**Cavus**	0	0	1 (100)	
	**Normal**	9 (20)	25 (55.6)	11 (24.4)	
	**Flat**	1 (12.5)	2 (25.0)	5 (62.5)	
		**Kappa index**	**CI 95%**		
	**Concordance**	0.167	−0.082	0.415	poor
	**Observed agreement**	56%			
**Right foot**		**Cavus**	**Normal**	**Flat**	
	**Cavus**	2 (66.7)	0	1 (33.3)	
	**Normal**	5 (11.1)	25 (55.6)	15 (33.3)	
	**Flat**	0	2 (25.0)	6 (75.0)	
		**Kappa index**	**CI 95%**		
	**Concordance**	0.253	0.019	0.487	poor
	**Observed agreement**	59%			
**STAHELI INDEX**					
**Total (both feet)**		**Cavus**	**Normal**	**Flat**	
	**Cavus**	51 (83.6)	6 (9.8)	4 (6.6)	
	**Normal**	17 (60.7)	3 (10.7)	8 (28.6)	
	**Flat**	8 (38.1)	3 (14.3)	10 (47.6)	
		**Kappa index**	**CI 95%**		
	**Concordance**	0.241	0.074	0.408	poor
	**Observed agreement**	58%			
**Left foot**		**Cavus**	**Normal**	**Flat**	
	**Cavus**	26 (86.7)	3 (10.0)	1 (3.3)	
	**Normal**	9 (56.3)	2 (12.5)	5 (31.3)	
	**Flat**	3 (37.5)	1 (12.5)	4 (50.0)	
		**Kappa index**	**CI 95%**		
	**Concordance**	0.258	0.019	0.496	poor
	**Observed agreement**	59%			
**Right foot**		**Cavus**	**Normal**	**Flat**	
	**Cavus**	25 (80.6)	3 (9.7)	3 (9.7)	
	**Normal**	8 (66.7)	1 (8.3)	3 (25.0)	
	**Flat**	5 (38.5)	2 (15.4)	6 (46.2)	
		**Kappa index**	**CI 95%**		
	**Concordance**	0.223	−0.012	0.458	poor
	**Observed agreement**	57%			

CI 95% = 95% confidence interval.

## Data Availability

Data sharing is not applicable to this article. The data are not publicly available because we must have to follow the ethical and legal guidelines.

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
