# Peer review of "Concordance between Pressure Platform and Pedigraph"

_diagnostics, 2021, doi:10.3390/diagnostics11122322_

Round 1
Reviewer 1 Report
This paper discusses an practical issue. The topic that has been conferred about in the paper is part of a comprehensive discussion on the problem of measuring reliability of devices for foot diagnostics.
While I think this is an interesting topic, the manuscript could be improved.
Abstract
Abstract needs editing. Contains redundant information, does not contain research methods, tools. The conclusions are incoherent.
Introduction
The Introduction section begins with specifying the structure and function of the foot.
The content in lines 34-37 needs to be supplemented. The structural features of the foot are also influenced genetic disorders such as Down's syndrome. Please discuss these issues in the introductory part by referring to the following article:
Puszczalowska-Lizis E, Nowak K, Omorczyk J, Ambrozy T, Bujas P, Nosiadek L. Foot structure in boys with Down syndrome. BioMed Research International 2017. ID 7047468: 6 pages. DOI: 10.1155/2017/7047468.
The study lacks of a clear pre-specified research questions.
Material and Methods
The Material and Methods section should begin with the section: Participants. Transfer to this chapter the fragment of Table 1 containing the following data: age, gender, body weight and height, BMI.
The inclusion and exclusion criteria require clarification.
What was the method of random selection? Or was this a convenience sample?
The study design, project description is not clear. Lines 92-93: The study of the plantar footprints was carried out through three measurements (Fig- 92 Figure 3) (11)……
Please specify.
After all, the chapter: Results does not include an analysis of the Clarke angle data. If so, also remove the drawing "A" from Figure 3.
However, did not give methods determining the length and width of the foot. Please complete in the figure.
Please describe / explain all analyzed indicators.
Was the consistency of the results with the normal distribution tested?
Line 126: (Portney LG, 2000) - assign a number.
Results
Please provide the norms based on which the foot categories (normal, flat, cavus) were distinguished according to the Chippaux and Staheli indexes. Provide references to the literature
Conclusions
Please specify the conclusions in points as answers to the research questions.
General comments to the Authors
That said, my comments are offered with the intent of helping the authors improve this manuscript. When the authors address these issues I will be able to comment definitively and make the final decision.
Author Response
Dear Rewiever,
I attach the comments that you suggest to us
Best regards,

Reviewer 2 Report
Thank you for the opportunity to review this paper. In general, I think it is a nice study, and the paper is decently written, but there is some room for improvement.
- Abstract. Please spend more space on describing the results. You don't have to tell about ethical approval in the abstract. Most importantly, tell about the poor concordance in the abstract, not only about the good results.
- Introduction could go into a little bit more depth. When should the mentioned indexes be used? Are they reliable in general? Have they been linked to injuries.
- I have similar issue with the discussion. Why could the poor concordance arise? What does this mean for clinical practice? What could be done differently to improve the concordance. In general, you focus too much on the positive aspect (correlation), and you almost completely avoid the negative aspect (poor agreement and reliability) in the discussion and conclusion (as well as abstract). These parts need to be amended.
Moreover, the paper should undergo an English proofread. It is not a huge issue, the paper was not difficult to read, but the English could be at least somewhat improved.
Author Response
Dear Rewiever,
I attach the comments that you suggest to us

Round 2
Reviewer 1 Report
The article has improved substantially.
Reviewer 2 Report
Thank you for addressing my comments. I have no further objections.